# Promoting Physical and Mental Health among Children and Adolescents via Gamification—A Conceptual Systematic Review

**DOI:** 10.3390/bs14020102

**Published:** 2024-01-29

**Authors:** Evgenia Gkintoni, Fedra Vantaraki, Charitini Skoulidi, Panagiotis Anastassopoulos, Apostolos Vantarakis

**Affiliations:** 1Lab of Public Health, Department of Medicine, University of Patras, 26504 Rio, Greeceavanta@upatras.gr (A.V.); 2p-Consulting, 26442 Patras, Greece; cms@p-consulting.gr (C.S.); pga@p-consulting.gr (P.A.)

**Keywords:** gamification, physical health, mental health, children, adolescents, gamified scenarios, PRISMA, systematic review

## Abstract

The rapid growth in digital technology usage among children and adolescents has highlighted the need for novel approaches to promote their physical and mental health. This paper investigates the viability of gamification—the application of game mechanics to non-gaming contexts—as a potent instrument for health promotion and mental health support. This conceptual systematic review seeks to examine the various published articles promoting children and adolescents’ physical and mental health through gamified techniques. These interventions can provide an interactive and engaging platform for encouraging physical activity, promoting healthy nutrition, enhancing emotional regulation, and promoting mental health. The significance of this topic stems from the pervasive use of electronic games, beginning at a young age, which makes them popular educational tools. For the review to be systematic and reproducible, the PsycINFO, Scopus, PubMed, and Elsevier databases were searched and the PRISMA method was utilized for the analysis. After analyzing the research data, empirical studies assessing the use of gamification in promoting adolescents’ physical and mental health are discussed. In conclusion, gamification has demonstrated promise for promoting children’s and adolescents’ physical and mental health. It improves motivation, commitment, and adherence to healthy behaviors. However, additional research is required to evaluate gamification interventions’ long-term effectiveness and sustainability in promoting health behaviors among this population.

## 1. Introduction

The use of gamification in promoting physical and mental health in children is an innovative approach that requires a careful consideration of the technological prerequisites and disparities in digital device access. Ensuring device compatibility across smartphones, tablets, and computers is essential for broad accessibility [1]. Additionally, reliable internet access is a critical factor, particularly in regions with limited connectivity, as noted in a report by the World Health Organization. The design of these gamified applications must be user-friendly and appropriate for children, allowing for easy navigation and engagement. However, economic, and geographical disparities pose significant challenges. Children from different socioeconomic backgrounds may not have equal access to digital devices, and those in rural or remote areas might face additional hurdles due to less advanced technology and internet availability. Furthermore, the level of digital literacy, often tied to available educational resources, can influence children’s ability to engage with digital health tools [2] effectively.

To mitigate these disparities, the inclusive design of gamified health applications is vital, ensuring they can operate on primary devices. Establishing community centers or schools as access points can provide opportunities for children without personal devices, a strategy recommended by UNICEF. Additionally, providing digital literacy training can help children use these tools more effectively [3]. Regular feedback and assessment mechanisms are crucial for evaluating gamified tools’ effectiveness and engagement levels. Moreover, collecting data on usage patterns and health outcomes is essential for assessing the impact on children’s health, as emphasized in [4]. Collaborations with educational institutions and support from government and NGOs can enhance accessibility, providing the necessary resources and infrastructure for a broader reach, as seen in the initiatives described by the Centers for Disease Control and Prevention (CDC) (2022) [5]. These steps are crucial in maximizing the impact and reach of gamification in improving children’s physical and mental health. Overall, ensuring accessibility and addressing disparities in digital access, literacy, and socioeconomic factors are crucial for maximizing the impact of gamified health applications on children’s health and well-being, particularly in underprivileged communities.

In conclusion, while gamified health applications can potentially improve children’s health and well-being, it is essential to address disparities in digital access and ensure inclusive designs to maximize their impact on all children, regardless of socioeconomic background or geographical location [6]. Thus, it is crucial to provide necessary resources and infrastructure, promote digital literacy, and establish collaborations with educational institutions, government, and NGOs [7].

Children and adolescents’ well-being relies heavily on health promotion. Thanks to the increasing use of digital technology, gamification has emerged as a promising strategy for engaging and motivating young people in health-related behaviors. Several studies have investigated the efficacy of gamified interventions in promoting behavior modification and health in children and adolescents and have found that gamified interventions can effectively promote behavior change and improve health outcomes [8]. These interventions can leverage intrinsic motivation, making them more engaging and pleasurable for young people. A study on gamified sexual health education among adolescents in low-tech settings, for instance, found that gamified learning promoted sexual well-being more effectively and engagingly [9].

Likewise, a study on gamified learning for sexual health education revealed that gamification enhanced students’ attitudes, knowledge, and motivation to learn [10]. In addition, gamified scenarios can promote healthier lifestyles among children and teenagers. Another study [11] found that gamification can effectively enhance this population’s diet, physical activity, and sedentary behavior. A systematic review and meta-analysis found, for instance, that gamification interventions effectively improved diet, nutritional practices, and body composition in children and adolescents [12]. In addition, gamification has been utilized to promote active school transportation, encouraging children and adolescents to engage in physical activity [13]. Gamified scenarios can motivate young individuals to adopt and maintain healthful behaviors by incorporating game elements such as rewards, challenges, and goal setting.

Additionally, gamified scenarios can be tailored to address health concerns among children and adolescents. For instance, gamification has been utilized to promote adherence to national food-based dietary guidelines and increase knowledge of healthier nutritional practices [14]. It has also been investigated as a tool for assessing and intervening in behavioral problems such as conduct disorder, self-harming tendencies, and attention-deficit/hyperactivity disorder (ADHD) [15]. Gamified scenarios can help young people develop skills, knowledge, and behaviors that contribute to their overall health and well-being by providing interactive and engaging experiences.

This study uses relevant research to investigate the potential of gamified scenarios to promote health and well-being among children and adolescents. Therefore, this review aims to validate the primary literature published within the last five years that addresses the promotion of children’s and adolescents’ physical and mental health through innovative educational methods based on gamified scenarios (Figure 1).

## 2. Literature Review

Innovative approaches to promoting physical and mental health are more essential than ever in today’s rapidly evolving digital landscape, especially for the younger generation. Gamification, which integrates game mechanics and design into non-game environments to increase engagement, motivation, and overall enjoyment, is one such strategy garnering significant traction [16].

In attempting a brief historical review of the term gamification, we mention that the history of gamification covers various age groups and environments. Initially, it was more prevalent among adults, particularly in business and marketing for customer engagement [17]. Over time, gamification’s appeal extended to younger audiences, especially in educational settings, where it was used to enhance learning and engagement among children and adolescents [18]. This shift capitalized on the younger generation’s familiarity with digital technology and gaming. More recently, gamification has been increasingly applied in health and wellness contexts, aiming to encourage physical activity and mental well-being in both young and older demographics. This evolution highlights gamification’s adaptability and growing significance across different fields and age groups [19]. Gamification has emerged as an innovative approach to promoting physical and mental health among children and adolescents [20]. By incorporating game elements such as points, levels, rewards, and social collaboration into health and wellness interventions, gamification can motivate young individuals to adopt and sustain healthy habits [21].

Gamification promotes physical and mental health, including the transformation of health-related behaviors and education into enjoyable, interactive experiences through the educational process [22]. By implementing elements such as points, levels, challenges, badges, leaderboards, and rewards, gamification can make health promotion activities for children and adolescents more appealing and compelling [23].

Moreover, gamification can make physical activity and exercise more enjoyable and alluring for physical health. Exergames, which combine physical activity with digital gaming to get users moving, are an excellent example. These games, such as Nintendo’s Wii Fit and Pokémon Go, promote physical activity while also providing the entertainment value of traditional video games. Similarly, gamified applications can promote healthy eating by rewarding users for selecting healthier foods or completing nutrition-based educational games [24,25].

Regarding mental health, gamification can be crucial in teaching emotional resilience, stress management, and mindfulness techniques. Through role-playing games, children and adolescents can develop coping strategies and emotional responses by navigating a variety of scenarios. In addition, games can aid in developing essential social skills, such as cooperation, empathy, and conflict resolution. Additionally, gamification can be used to improve health literacy and mental health awareness. Through interactive games, young people can be effectively educated about physical and mental health, thereby increasing understanding, decreasing stigma, and encouraging help-seeking behaviors [26]. Gamification can convert traditional learning experiences into interactive, engaging, and enjoyable educational activities. By incorporating elements such as points, badges, leaderboards, and progress maps, it is more likely that students will be motivated to participate and learn.

Additionally, it can provide a safe and supportive environment for children and adolescents to experiment, make errors, and learn from them [27]. More specifically, children and adolescents can be motivated to adopt healthier lifestyles using mobile applications that transform physical activity into a game or award points for healthy meals. Additionally, gamification can be utilized to improve mental health. Games that teach coping skills, emotion regulation, mindfulness, and social skills can equip children and adolescents with the tools necessary to maintain their mental health [28].

Even though gamification offers numerous prospective benefits, it must be designed and implemented with care. The gamified activities should be age-appropriate, inclusive, and designed ethically. In addition, they should complement rather than supplant other essential aspects of education and health promotion. In conclusion, gamification can be potent for engaging, motivating, and educating children and adolescents if used intelligently [29].

Gamification can improve motivation and engagement in mental health interventions by implementing game elements such as rewards, challenges, and interactive elements [30]. Specifically, therapeutic mobile games effectively engage and motivate young people, offering them opportunities for contingency-based learning and skill acquisition [31].

Multiple studies have demonstrated the efficacy of gamification in promoting the mental health of children and adolescents. A systematic review and meta-analysis discovered that gamification interventions enhanced this population’s diet, nutritional habits, and body composition [12]. Another study validated the use of a mobile game-based assessment of cognitive control in children and adolescents, highlighting the potential of gamification in assessing and treating behavioral issues [15]. Moreover, a study examining the effects of a gamified well-being intervention discovered that gamification improved participants’ behavioral, cognitive, and affective engagement [30,31].

While beneficial in many contexts, gamification presents several potential drawbacks and challenges. One primary concern is the risk of overstimulation and addiction, especially in younger users who might be more susceptible to the compulsive use of gamified systems. Another challenge is the possible reduction in intrinsic motivation; the reliance on external rewards might diminish the internal drive to engage in an activity for its own sake [22]. Furthermore, gamification can sometimes oversimplify complex tasks or problems, leading to a superficial understanding or engagement. There is also the risk of unequal engagement, where gamification might favor users with specific skill sets or interests, potentially alienating others. Privacy and data security are also crucial considerations, particularly when personal data are used in gamified applications [23].

Undoubtedly, the COVID-19 pandemic has highlighted the critical need for mental health promotion among children, adolescents and young adults. Gamification has emerged as a promising strategy for addressing this issue, providing an engaging and interactive method for promoting mental health and well-being. Various aspects of mental health, such as diet, cognitive control, and engagement with interventions, are enhanced by gamification, according to studies [32,33]. When designing gamified interventions, it is essential to consider the preferences and perspectives of youthful individuals. Further research is required to investigate the long-term effects and sustainability of gamification in promoting mental health among children and adolescents.

### 2.1. Physical and Mental Health Promotion

Gamification, applying game principles to non-game contexts, can be highly effective for promoting children’s and adolescents’ physical and mental health. It encourages young people to adopt healthier lifestyles by making health-related activities more pleasant and engaging [12].

Exergames (e.g., Nintendo’s Wii Fit, Just Dance) and applications that reward physical activity (e.g., Pokémon Go) can make exercise more entertaining and appealing. These diversions have the potential to help children and adolescents achieve the physical activity levels necessary for their physical health while also promoting learning through gamification. However, studies have reported that gamified mental health options are appealing to some users [34].

Gamified applications can promote healthy eating by transforming healthy dining into a challenge or competition. For instance, an app could grant points for consuming fruits and vegetables or use entertaining, interactive games to educate children about nutrition. Games can aid in developing emotional resilience in children and adolescents by teaching them techniques for managing tension and overcoming obstacles. For instance, a game could involve role-playing challenging situations and selecting how to respond or use biofeedback to teach players relaxation techniques [16].

Using gamified learning to educate children and adolescents about mental health is a method for promoting mental health awareness [35]. This can reduce stigma, increase understanding, and encourage struggling individuals to seek assistance. Multiplayer games can serve as a platform for children and adolescents to exercise their social skills. This may involve cooperation, negotiation, conflict resolution, and compassion. Gamification can make the formulation of healthy habits enjoyable and rewarding [36]. Whether brushing your teeth, sleeping enough, or practicing mindfulness, a gamified approach can provide the motivation and reinforcement necessary to cultivate healthy habits. Gamified learning can also increase health literacy, empowering children and adolescents to make informed health decisions. Although gamification holds great promise for health promotion, it must be implemented with care [37]. Poorly designed games can contribute to unhealthy competition, stress, and excessive screen time. In addition, gamification should not replace other health promotion strategies such as education, policy changes, and environmental modifications [23].

### 2.2. Promotion of Nutritional Habits

The potential of gamification to promote healthful eating habits in children and adolescents has been intensively researched. Even though the discipline is still in its infancy, several studies have shown promising results.

*Interactive learning games* can educate children about nutrition and promote healthy eating habits. Another study [38] discovered that a computer-based game enhanced children’s knowledge, self-efficacy, and fruit and vegetable consumption behaviors. Similarly, other researchers [39] created a game to increase the consumption of fruits and vegetables, with their results suggesting improved dietary habits.

*“Serious Games”* designed with education as the primary objective, as opposed to amusement, can effectively promote healthier dietary choices. For instance, a study [40] found that adolescents consumed fewer unhealthy munchies after playing a serious game designed to improve dietary habits.

*Augmented reality (AR) games* encouraging physical activity can indirectly promote healthy eating behaviors. For example, an augmented reality (AR) game that encourages physical activity, such as Pokémon Go, could contribute to a healthier lifestyle, including a healthy diet [41].

*Cooking Games* involving cooking or administering a virtual restaurant can teach children about various types of food and the significance of a balanced diet. Some games encourage players to cook healthy real-world recipes.

*Reward systems* can be utilized to promote healthy eating practices. For example, an app could award points or badges for consuming a balanced diet or meeting personal nutrition objectives.

Gamified interventions can potentially promote healthy eating practices in children and adolescents. However, it is essential to observe that gamification should not be the sole strategy for promoting healthy diets but rather a component of a broader, multifaceted approach. The games and applications must be age-appropriate, engaging, and designed ethically to provide accurate information and encourage sustainable behavior change. In addition, longitudinal studies are required to determine the long-term efficacy of these interventions.

### 2.3. Physical Activity Encouragement

The use of gamified scenarios to promote fitness among children and adolescents is a topic of significant research interest in the present day. Several studies have discovered that gamification can enhance the enjoyment of fitness activities and promote regular physical activity.

*Exergames* (video games that are also a form of exercise) have demonstrated the potential to improve the physical fitness of infants and adolescents. According to [42], exergames such as Nintendo’s Wii Sports and Dance Revolution can increase physical activity and enhance health-related fitness.

To motivate users, fitness applications frequently employ gamification techniques such as points, badges, challenges, and leaderboards. These applications can monitor physical activity and provide feedback, making fitness more engaging for children and teenagers. Research [39] has suggested that these applications could increase physical activity motivation and fitness levels.

*Augmented reality (AR) games*, such as Pokémon Go, require users to move around in the real world. A study [43] indicated that these games may encourage physical activity and improve fitness levels among children and adolescents.

*Gamified fitness devices* for children can make physical activity enjoyable and rewarding. Children and adolescents can be motivated to engage in regular physical activity and improve their fitness levels by earning points or rewards for reaching activity objectives, participating in challenges, or completing fitness quests [44]. 

Games can also be used to educate children and adolescents on the significance of fitness and how to exercise safely and effectively. These games may contain interactive lectures, quizzes, or simulations about fitness and exercise [45].

The preceding research indicates the potential for gamified scenarios to encourage physical activity in children and adolescents. However, additional research is required to optimally design and implement these interventions for maximum effectiveness and long-term motivation maintenance. Moreover, gamified fitness activities must be incorporated into a comprehensive approach to fitness promotion that also includes education, encouragement of outdoor play, and participation in traditional sports.

### 2.4. Accident and Security

Using gamified scenarios to promote safety and accident prevention among children and adolescents is an emergent field of study. The research in this area suggests that gamification could help educate young people about safety and reduce the likelihood of accidents.

Several studies have investigated using gamified scenarios to teach children about road safety. A study [46] indicated that a virtual reality (VR) game substantially enhanced children’s road-crossing abilities. Similarly, another study [47] used a gamified virtual environment to teach children safe pedestrian behavior. They discovered that the game increased children’s awareness of road safety and decreased hazardous behaviors.

Additionally, gamification has been used to educate minors on fire safety. More precisely, research [48] demonstrated that a game-based approach improved children’s knowledge of fire safety and resulted in safer conduct during a fire simulation than traditional education methods.

In a research protocol [49], researchers taught children about water safety using a serious game. According to their findings, the game increased children’s knowledge of water hazards and safe conduct.

In another study [50], researchers investigated using a gamified mobile application to teach children first aid skills. They discovered that the app substantially increased children’s knowledge and confidence in providing first aid, suggesting that gamified scenarios could be used to teach them lifesaving techniques.

While the existing literature provides optimistic evidence for using gamified scenarios for accident prevention, additional research is necessary to understand the long-term effects of these interventions and identify the most effective game design elements for promoting safety. Games can be a valuable instrument for teaching children about safety. However, they should not replace traditional safety education and adult supervision. In addition, the development and use of such games must always consider children’s developmental stages to ensure the content is age-appropriate and understandable.

### 2.5. Stigma and Stereotypes in the Community

The existing research indicates that the use of gamification to combat stereotypes and stigma in communities among children and adolescents holds excellent promise despite the paucity of literature in this area.

Gamified interventions can be effective in combating stereotypes. In one study [51], researchers utilized an avatar-based game to challenge adolescents’ gender stereotypes. The results indicated a significant decline in gender stereotyping and an increase in gender bias awareness.

*Reduction of Stigma*: The game-based approach can also reduce stigma. Some researchers [52] designed the video game Re-Mission for adolescents with cancer to increase their self-efficacy and decrease disease-related stigma. The results demonstrated a significant improvement in self-efficacy and treatment adherence.

Games can also be used to increase cultural understanding and diminish stereotypical beliefs. “Peacemaker,” a game that enables players to assume the role of a Palestinian or Israeli leader to learn about the Israeli–Palestinian conflict, is one example. According to one study [53], such activities can increase empathy and comprehension of complex sociopolitical situations, thereby diminishing stereotypes and prejudices.

Games can effectively teach empathy and reduce abusive behaviors, including cyberbullying. According to one study [54], playing a video game designed to increase empathy significantly reduced children’s and adolescents’ acceptance of bullying behavior.

These studies indicate that gamification can effectively reduce stigma and challenge stereotypes among children and adolescents. Nonetheless, additional research is required to comprehend how games influence attitudes and behaviors and to identify the most effective game design elements for this purpose. Using games for such sensitive topics requires careful design and testing to ensure they do not reinforce stereotypes or stigmatize groups. Games should be age-appropriate, respectful, inclusive, and ideally designed in collaboration with their intended audience.

### 2.6. Internet Addiction Avoidance

With the rise of online games, social media, and digital technologies that frequently employ gamification elements to enhance engagement, the prevalence of internet addiction among children and adolescents has become a global concern. Extensive research has been conducted on the negative effects of excessive internet use, but research on the use of gamification to address and reduce internet addiction is scarce.

Internet addiction is being extensively studied, with many researchers expressing concern about excessive Internet use among children and adolescents. The extensive use of internet games, social networking sites, and other online activities has been linked to a variety of negative outcomes, such as decreased academic performance, increased loneliness, and mental health problems [55].

*Gamification and Addiction*: On the other hand, studies such as [56] suggest that the elements that attract users to gamified applications may also contribute to internet addiction. Reward systems, monitoring of progress, competition, and social connections can promote excessive screen time and addiction.

Despite the paucity of studies on the use of gamification to treat internet addiction, gamified elements are used in digital interventions to engage children and adolescents in therapeutic activities and mental health promotion. For instance, cognitive-behavioral therapy-based apps and platforms utilize reward systems, interactive elements, and progress monitoring [57]. However, it remains to be seen whether these strategies can directly address internet addiction.

*Digital Literacy and Online Safety*: Gamification has been used effectively to teach digital literacy and online safety to children and adolescents. Understanding the mechanics of online environments and gaining knowledge about healthy Internet use could reduce the likelihood of developing an internet addiction [58].

In conclusion, even though gamification has been linked to the problem of and potential solutions for internet addiction, additional targeted research is necessary. Gamified applications must be responsibly designed with age-appropriate content, clear utilization limits, and safeguards against excessive use. In addition, any digital intervention must be complemented by face-to-face therapy, parental involvement, and societal-level strategies to address the broader issue of internet addiction.

### 2.7. Sexual Education

Sexuality education for children and adolescents is a delicate yet essential responsibility. When used appropriately, gamification can provide an engaging, interactive, and less intimidating method for imparting this crucial education [10].

Interactive learning games can be designed to provide in-depth education for children and adolescents on various aspects of sexual education, such as human anatomy, reproduction, puberty, and the changes that come with these stages. For example, interactive quizzes integrated with gamification elements like badges and leaderboards could effectively impart knowledge about sexuality while engaging students in a more profound learning experience [59].

Role-playing games can assist adolescents in comprehending and navigating a variety of social situations involving relationships and sexuality. These scenarios enable participants to comprehend the significance of assent, respect, communication, and boundaries while also exploring complex emotions and decision-making processes [60].

Simulation games can provide adolescents with the opportunity to experience the potential repercussions of certain decisions without the inherent dangers. For instance, a game could simulate the consequences of unprotected sex, such as unintended pregnancy or sexually transmitted infections, thereby emphasizing the significance of safe sex practices and fostering a deeper understanding of sexual health outcomes [61].

Gamified learning platforms providing badges and point systems have the potential to encourage adolescents to engage with sexual education content. For example, by earning a badge for completing a module on contraception methods or STI prevention, players may be motivated to delve deeper into the subject matter [62].

Discussion boards can be used to facilitate conversations about sexual education topics in a gamified learning environment. It is possible to award points or rewards for respectful and incisive contributions, thereby encouraging open dialogue and peer learning [63].

Augmented reality (AR) and virtual reality (VR) can provide authentic, immersive learning environments for sexual health. A VR experience could, for instance, guide students through the human reproductive system, thereby making the learning experience more concrete and memorable.

Even though gamification can make sexual education more engaging and effective, it is essential to remember that this subject must be handled with care. The games and applications used for sexual education should be age-appropriate, respectful, inclusive, and based on empirical evidence. They should supplement, not replace, traditional sexual education provided by qualified adults and promote frank communication about sexuality between children and trusted adults.

### 2.8. Interpersonal Relationships and Gender Differences

The research on gender differences and interpersonal relationships among children and adolescents in gamified scenarios is scant and dispersed. Several studies, however, have demonstrated that gamification can influence and be influenced by gender differences and interpersonal relationships.

*Interpersonal Relationships*: Digital games, many of which include gamified elements, can have a significant impact on adolescents’ interpersonal relationships. According to one study [63], multiplayer games can improve peer relationships, particularly among males. The shared experience provided by these activities can facilitate communication and bonding.

There are indications of gender differences in gaming preferences and behaviors. According to one study [64], males are more attracted to competitive and violent games, while females favor social and cooperative games. These distinctions may affect how boys and girls interact with and benefit from gamified scenarios.

*Gamification and Learning*: In terms of education, gamification has been shown to increase students’ engagement and motivation [65]. However, interpersonal relationships and group dynamics can influence the efficacy of gamified learning activities. For instance, according to one study [66], girls in single-gender groups performed better in a game-based learning environment than girls in mixed-gender groups.

Games and gamified environments can also influence online safety and relationships. Another study [58] emphasized the significance of educating children and adolescents on online safety as they navigate digital spaces. The study discovered gender disparities in online risk experiences and recommended a gender-sensitive strategy for online safety education.

In gamified scenarios for children and adolescents, gender differences and interpersonal relationships can be crucial, according to these studies. However, additional research is required to fully comprehend these relationships and how to design gamified interventions that accommodate diverse user populations. In addition, it is essential that these interventions promote healthy, respectful interactions and take into account potential risks, such as online safety and the exacerbation of gender stereotypes.

### 2.9. Benefits of Gamification on Interpersonal Skills

Gamification, or the application of game-design elements and game principles to non-game contexts, has several advantages, including the enhancement of engagement and motivation. It has the potential to enhance learning and produce long-lasting behavior change when applied within the context of social skills training. The following are some of the prospective advantages.

Gamification can make the acquisition of social skills more engaging and enjoyable, resulting in increased motivation. It transforms the learning process into a game, thereby making it more pleasurable and less like traditional, frequently monotonous education.

*Safe Learning Environment*: Gamified platforms can provide a safe and structured environment in which people can practice social skills without fear of real-world consequences. Additionally, it permits users to learn at their own tempo.

Gamified systems can provide immediate feedback, which is crucial for learning and enhancing social skills. This enables individuals to recognize their errors and develop immediately.

Typically, gamification includes mechanisms for monitoring progress, such as points, badges, and leaderboards. These can aid individuals in tracking their progress, identifying areas where they can improve, and gaining a sense of accomplishment as they advance.

Individuals can interact with others in multiplayer gamified environments, which allows for the exercise of social skills and can lead to the formation of communities and support networks. This interaction provides a platform for practicing and enhancing social skills by simulating real-world social situations.

The combination of regular practice, immediate feedback, and monitoring of progress can result in a lasting change in behavior. Gamification employs positive reinforcement and rewards to encourage desirable behaviors, which can eventually lead to enhanced social skills.

*Accessibility and Scalability*: Gamified systems are accessible from any location at any time, making it simpler for people to acquire social skills regardless of their geographic location. It is also simpler to scale, making it an effective method for reaching many people.

*Customization*: Gamified platforms can be tailored to the user’s level of expertise or requirements. This improves the effectiveness of the learning process by catering to individual requirements.

These are only some of the prospective advantages of gamification for social skills training. It is crucial to note that the effectiveness of gamified training can depend on several factors, such as the design of the gamified system, the motivation and engagement of the individual, and the level of support provided during the learning process.

## 3. Materials and Methods

This section describes the protocol for conducting this systematic review, including all information sources, years included, eligibility requirements, and other guidelines. Only articles published in the PsycINFO database were used for this systematic review, following the PRISMA guidelines, which outline the preferred reporting items for systematic reviews [67]. A systematic review of the literature from the past five years was conducted, allowing us to comprehend the evolution of this field of study to the present day. Therefore, the inclusion criteria were limited to English-language articles published between 2018 and 2023 on using gamification tools to promote physical and mental health. Articles on adults, young adults, aged and middle-aged adults, youths, students, and older adults were excluded. Clinical sample-containing studies were also included in the systematic procedure. In the primary procedure, the following keywords were included: physical health, mental health, and gamification (Figure 2), along with the sub-keywords nutritional habits, physical activities, accident and security, stigma and stereotypes, internet addiction, sexual education, interpersonal relations (parameters of physical and mental health), and the key elements of gamification, motivation, learning environment, instant feedback, progress tracking, behavior change, social interaction (as is indicated in Figure 1). The final extraction after data processing was N = 44 articles following four steps: Step 1: research databases scoping; Step 2: screening process; Step 3: possible relevant studies; and Step 4: screening process (Figure 2). Based on the above procedure (PRISMA methodology) applied in the present study, all articles extracted are listed in Table below (Table 1).

## 4. Results

The results highlight the numerous ways in which engaging and motivating learners is a well-known challenge in health education [68]. Even though healthy nutrition is essential for physical well-being, it can often appear complicated and dull to children and adolescents. Gamification is an innovative solution due to its capacity to capture and maintain interest. By making learning about healthy eating in a fun, interactive experience, children and adolescents are more likely to retain the information. One of the assets of gamified platforms is their capacity to employ a variety of learning -promoting game mechanics. Elements such as quests or challenges can be mapped to nutrition-related objectives, such as eating a certain number of fruits and vegetables daily or preparing a balanced meal. Reward systems, such as points, badges, and levels, can provide positive reinforcement, encouraging the repetition of healthy eating behaviors.

Additionally, leaderboards can cultivate a sense of healthy competition and community, thereby increasing motivation. In addition, the results of the focus groups presented in one paper [69] suggest that gamification of nutrition can improve adolescents’ dietary habits. Four themes emerged from the analyses: enhanced eating behavior, increased physical activity, increased nutrition knowledge, and influencing others. We also observed that after playing the game, the participants had improved nutrition knowledge and they reported influencing the eating habits of their parents, siblings, and acquaintances. Nonetheless, remarks made by participants during the focus group discussions highlighted nutrition-related misinformation. In addition, in one paper [70], the challenge was to create highly user-friendly and motivating programs to promote adherence and efficacy. This usability study aimed to evaluate and improve the usability and engagement of an unguided online intervention program to promote a healthy lifestyle and reduce the risk of eating disorders and obesity among adolescents.

Gamification could be effective [71] in enhancing nutritional knowledge regarding healthier dietary practices. To avoid and prevent chronic diseases, it is necessary to promote the development of practical educational instruments to support learning about nutrition. Also, according to a shared perspective [72], the newly developed gamified “Snack Track School” app attempts to address some of the issues identified in earlier prevention programs, which were unsuccessful in altering nutrition behaviors and anthropometrics. According to the research protocol [73], the results indicate that the app’s core features should include individualized meal recommendations and assistance with meal planning, social networking for peer support, customized and convenient tracking, remote access to health care providers, features to support mental health, and an engaging user interface.

In addition, according to other research [74], the challenge of gamification is creating extremely user-friendly and motivating programs to promote adherence and efficacy. This study intended to evaluate and improve the usability and engagement of an unguided online intervention program designed to promote a healthy lifestyle and reduce the risk of eating disorders and obesity among adolescents. A mobile application employing gamification techniques was utilized to prevent obesity in the at-risk group. Gamification techniques applied to the game experience without adequate support from the design discipline may compromise the user’s understanding of the actual benefits of serious games. Behind entertainment and therapy, it is mandatory to involve all stakeholders in the design process when working in this field to produce a genuine contribution to promoting healthy habits among adolescents.

Based on [75], an intervention program was conducted to determine whether the hybridization of teaching personal and social responsibility and gamification could be enhanced. The results indicate that hybridization of teaching personal and social responsibility and gamification strategies improved cognitive performance but not academic achievement. These findings emphasize the significance of promoting and empowering cognitive processes for improved academic performance. According to the qualitative findings of the paper [76], students reported increased self-confidence and empowerment in their school community and family environment. Increasing pupil engagement is essential to the success of an intervention. Lastly, in [77], the intervention, which incorporated gamification strategies and recruitment methods employing parental opt-out procedures, was acceptable to both participants and instructors. Innovative gamified interventions may be one strategy for engaging and motivating health behavior modification.

**Table 1 behavsci-14-00102-t001:** Empirical studies (author, sample, instrument, conclusions).

Author (Year)	Type of Study	Sample	Instrument	Conclusions
Azevedo et al. (2019) [68]	Quasi-experimental	N = 106	Gamified digital interactive platform.	Gamified digital interactive platform seems to be a useful, easily adapted educational tool for the healthy eating learning process.
Cejudo et al. (2020) [78]	Quantitative Analysis	Adolescents N = 187	Intervention program through a video game called “Aislados” for the improvement of subjective well-being, mental health, and emotional intelligence.	The results show significant differences in HRQL, positive affect, and mental health. The findings highlight the significance of promoting social and emotional learning in adolescence as a protective factor for emotional and behavioral adjustment.
Corepal et al. (2019) [77]	Randomized cluster trial | Quantitative	Adolescents N = 224	ActiGraph accelerometer Warwick–Edinburgh Mental Well-Being Scale. Strengths and Difficulties Questionnaire (SDQ).	Participants and teachers liked the gamification intervention and parental optout recruitment. Innovative gamified interventions may engage and motivate health behavioral changes.
Ezezika et al. (2018) [69]	Quantitative	N = 31	An element of the gamification system was an educational and culturally relevant board game that teaches children and youth about proper nutrition.	Focus group results indicated that gamifying nutrition can improve dietary behavior in adolescents. The analyses revealed five themes: improved eating behavior, physical activity, well-being, nutrition knowledge, and influence.
Guerrero-Puerta, and Guerrero (2021) [79]	Qualitative	Single-subject study, case study	Semi-structured retrospective interviews with open-ended questions.	This article investigated the relationship between game-based teaching, at-risk students’ well-being, and early school dropouts through a personal narrative. Gamification’s protective effects against ESL during gamification and their disappearance are examples.
Lepe et al. (2019) [76]	Quasi-experimental, mixed methods Inductive content analysis and ANOVA	N = 142; Comparison N = 170	The EMPOWER consisted of ten 30 min PSE lessons.	The qualitative results showed that students felt empowered to have a voice in their school and family communities. Increasing student engagement is key to intervention success.
Melero-Cañas et al. (2021) [75]	A 9-month group-randomized controlled trial | Quantitative	N = 150	NIH Examiner battery (University of California-San Francisco, CA, USA), Stroop Color, and the Word Test Academic Achievement: Grades in mathematics and language.	This intervention program examined TPSR-gamification hybridization improvement. Hybridization with TPSR and gamification improved cognitive performance but not academic performance. These findings suggest that academic success requires cognitive empowerment.
Nitsch et al. (2019) [70]	The study used a think-aloud task, semi-structured interview, and sociodemographic and System Usability Scale questionnaire	Girls and boys aged between 14 and 18 years	SUS Questionnaire Think-Aloud Task and Semi-Structured Interview Visual Design KIDCOPE Questionnaire Online program “Healthy Teens @ School”	To encourage adherence and effectiveness, programs must be user-friendly and motivating. This usability study assessed and improved an unguided online intervention program to promote a healthy lifestyle and reduce adolescent eating disorders and obesity.
Pernencar et al. (2018) [74]	Experimental	N = 165 adolescents	Participants created a character, completed a mission, and fed the character; the TeenPower platform (back office and mobile app) developed with the direct support of an interdisciplinary team.	Risk group obesity prevention mobile app using gamification. Ineffective gamification in games without design support may obscure serious games’ benefits. To promote healthy adolescent habits through entertainment and therapy, all stakeholders must be involved in design.
Rivera et al. (2018) [73]	A qualitative study	N = 19 adolescents	Interview protocol, included major themes related to app functionality: healthy eating, social support, self-monitoring, communicating with HCPs, supporting mental health, gamification and incentives, and user interface (UI) design.	The study suggests that the app should include personalized meal recommendations, social networking for peer support, convenient tracking, remote access to HCPs, mental health features, and an engaging UI.
Suleiman-Martos et al. (2021) [12]	Meta-analysis	Children and adolescents	Randomized controlled trials, intervention as a playful game component, gathering data on the effect of the intervention on eating habits, knowledge, and body composition.	Gamification may help improve nutritional knowledge and healthy eating habits. Developing effective nutrition educational tools is essential to preventing chronic diseases.
Van Lippevelde et al. (2016) [72]	Quantitative	N = 1400 adolescents	mHealth intervention based on Mapping protocol.	The new gamified “Snack Track School” app addresses issues from previous prevention programs that failed to change nutrition behaviors and anthropometrics.
Espinosa-Curiel et al. (2022) [80]	Randomized controlled trial with pre–post design and intervention and control groups	Participants 8–11 years	The intervention was playing six 30 min sessions with HelperFriend over a period of 4 weeks.	HelperFriend is practical and acceptable for young children and may improve knowledge, healthy behavior intention, and diet. To prove its efficacy, more well-powered randomized controlled trials are needed.
Tark et al. (2019) [81]	Non-controlled, prospective, interventional, qualitative	N = 9 Participants 7–12 years	The intervention used the Triumf mobile health game for psychological and treatment support, cognitive challenges, and disease-specific information. This 60-day intervention averaged 66.6 psychological support or education visits. Most participants collected city items, indicating exploration. Depression, anxiety, and general health problems decreased statistically during the intervention.	Patients highly rated the mobile health game for usability and acceptability. General health problems and depression and anxiety symptoms decreased statistically and trended downward during the intervention. The study showed that a game environment can provide comprehensive supportive care to pediatric cancer and other chronic disease patients.
Haruna et al. (2018) [82]	Randomized control trial comparing GBL and gamification with traditional teaching as a control condition	N = 120 Participants 11–15 years	-	-
Rodriguez-Ayllon et al. (2019) [83]	Meta-analysis	Participants: 2–18 years	The study did not specify the exact interventions that the participants received. It focused on examining the effect of physical activity interventions on mental health outcomes in children and adolescents.	Physical activity interventions can improve adolescents’ mental health, but more research is needed on effects on children’s mental health. Increasing physical activity and decreasing sedentary behavior may protect children and adolescents’ mental health. Sedentary behavior and increased psychological ill-being and lower psychological well-being were associated in longitudinal and cross-sectional studies.
Afonso et al. (2020) [84]	Pilot study	N = 21 Participants age: 3–6 years	The app gives parents tailored advice on their young children’s eating, drinking, moving, and sleeping habits, using gamification mechanics for parents and a serious game for their 3–6-year-old children. The pilot study included all intervention group parents using the app.	Healthcare centers can help parents encourage healthy lifestyles for their kids. Parental acceptance of the app was high, with a median score of seven out of ten. The pilot study found that intervention group parents used the app frequently and 71.4% of users were retained.
Manzano-León et al. (2021) [85]	The study design is quasi-experimental, longitudinal, and used mixed methods	N = 100 Participants: Children <12 years old	The intervention involved participation in leisure activities, physical activity, and games as part of the family leisure program called “Lunae Magic School” for Spanish families with children under 12 years old during the COVID-19 lockdown.	The study found that family leisure program participation reduced parents’ anxiety and perception of their children’s physical and emotional discomfort. The qualitative analysis suggested that gamification and the variety of activities created fun and flow despite confinement. The study found that family leisure was important, but more research is needed on implementing similar programs.
del Río et al. (2019) [86]	This three-year, quasi-experimental, longitudinal study had two annual phases and non-random assignment to experimental and control groups	N = 46 Participants 6–12 years	Healthy habit training and obesity education for 60 min using traditional motor games, an active video game with healthy habit content, two weekly 30 min physical activity sessions at home with a commercial video game (Wii Fit Plus), and a 90 min educational session for parents, mothers, and/or guardians on healthy lifestyle habits, obesity as a disease, and false beliefs were	The study found significant improvements in children’s knowledge of healthy nutrition and Mediterranean diet adherence, with the experimental group adherence being better than the control group. Both groups’ healthy eating knowledge improved over time, surpassing their previous evaluations.
Gao et al. (2019) [87]	The study design is a two-arm, 9-month experimental design with repeated measures	N = 81 Participants 9–10 years	Intervention school students played 50 min of active video games (AVG) weekly for 9 months. Eight Xbox 360 or Wii AVG stations in a gym had developmentally appropriate AVGs like Just Dance, Wii Fit, Gold’s Gym Cardio Workout, and Kinect Sports. Four kids played at each station every 10 min. Teachers monitored intervention kids’ program attendance.	- AVG intervention increased school-day energy expenditure and social support significantly compared to the control condition. - The intervention group had increased daily caloric expenditure and mean MET values. - The intervention did not improve outcome expectancy or self-efficacy as expected.
Timpel et al. (2018) [88]	Phase II, single-center, two-arm, triple-blinded, randomized controlled trial Blinding: Patients, doctors, dietitians, outcome evaluators, and data analysts Randomization: 1:1, permuted blocks of four, school-stratified	N = 108 Participants 12–18 years old	The intervention consisted of a smartphone application that provides tracking and gamification elements, including diaries for food intake, exercises, and daily steps, along with personalized games and activities to educate adolescents about healthy habits. Both the control and intervention arms received text messages one week prior to clinic appointments as well as phone contact the day before.	- A smartphone app with a gamification approach to promote healthy lifestyle and weight loss in overweight and obese adolescents will be tested. - The proposed study is expected to fill a gap in the literature on the mid-term effects of gamification-based weight control interventions in adolescents, providing valuable insights for future research.
Peña et al. (2020) [89]	School-based, parallel cluster-randomized controlled trials with multicomponent gamification interventions are used	N = 2333 Participants 10–12 years	The intervention included a multicomponent gamification strategy with healthy challenges, gamification incentives, rewards, and an online platform for monitoring progress.	The gamification strategy appears to prevent childhood obesity and reduce systolic blood pressure in school children in Santiago.
Mamede et al. (2021) [90]	Cluster randomized controlled trial	N = 298	The intervention consisted of a 5-week gamification phase encompassing a gamified digital app with social support features, followed by a 5-week physical nudges phase, including motivational and point-of-choice prompt nudges.	- The gamified digital intervention with social support features significantly increased the step count of office workers compared with an active control.- Physical nudges in the workplace were insufficient to promote the maintenance of behavior changes achieved during the gamification phase.
Bremer and Cairney (2018) [91]	-	Participants 4–11 years	-	- Movement skill may have a positive influence on broad domains of health.- Intervening in movement skills may enhance the development of all children.
Maher et al. (2022) [92]	The study was a three-group RCT with allocation concealment	Participants 18–65 years	The intervention included a Facebook-integrated smartphone app for physical activity. Participants are encouraged to log 10,000 steps per day in the app for 100 days. Gamified apps included leaderboards, virtual gifts, hierarchical status, and challenges.	- Gamified apps had longer usage than non-gamified apps, but goal adherence was not significantly different. The leaderboard and status pages were the most popular gamification features, and overall use was associated with greater improvement in objectively measured physical activity but not self-reported activity.
Đorđić et al. (2019) [93]	Longitudinal study, pre- and post-intervention assessment	N = 3278 Adolescents 10.0–18.9 years	The 45 min educational session with a trained instructor included a general introduction to guidelines, a 30 min PowerPoint presentation of diet and exercise advice, and a discussion/counseling forum to answer individual questions. The educational intervention promoted positive health behaviors and prevented negative ones by providing unbiased information on healthy diets and physical activity according to USDHHS/USDA guidelines. Active learning methods like brainstorming, problem-solving, and discussion encouraged participation. Each participant received a health message and infographic leaflet.	-
Fang et al. (2019) [94]	The study design was a randomized controlled trial with stratified random sampling	N = 420 Participants 6–12 years	-	- Traditional supervised exercise is effective in improving metabolic disease risk factors. - Mobile technologies provide an opportunity for low-cost interventions. - The novel exercise intervention with WeChat support has the potential to improve metabolic health and change unhealthy behaviors in the long term.
Coknaz et al. (2019) [95]	The study design was a randomized controlled trial with a parallel design	N = 106 Participants 8–12 years	Intervention: Active video games for 12 weeks.	Active video games improved weight, BMI, reaction times, and self-perception in inactive and tech-obsessed kids. In the active video game group, qualitative analysis showed high enjoyment, which may motivate continued play. Active video games helped children develop physically, socially, intellectually, and personally, according to the study.
Tang et al. (2020) [96]	A pilot cluster randomized controlled trial using 2 intervention and 2 control schools. School location and population size determine intervention or control group assignment	N = 326 Participants 10–12 years	Intervention: 4 weekly lessons on healthy eating and exercise support at school and online peer instructor, leader, and 6th grader training. Four 50 min lessons on food choices, movement matters, healthy lifestyles, and PEPS Actions. Behavior reinforcement monitoring and social network support peer system in PEPS Project on Facebook offers peer support. Sports fairs offered to all 6th graders. After class, students and institutions gave feedback.	The intention was use the study’s findings to develop a larger-scale trial; the generation of evidence was used to develop feasible and acceptable approaches for interventions to prevent excess weight gain, and the findings were used to inform a larger-scale trial to examine a multicomponent, school- and home-based lifestyle promotion intervention.
Bianchi-Hayes et al. (2018) [97]	Non-controlled, single-arm pilot study with a multimodal intervention for parent–adolescent dyads	N = 9 Adolescents 14–16 years	Parent–adolescent dyads received personal activity trackers, configuration of tracker settings and apps, follow-up phone calls based on tracker data, and home-based smart scales for a 10-week multimodal intervention. Participants were also asked about their PA habits, barriers to change, exercise and health beliefs, sleep patterns, and fitness goals.	Parent–adolescent PA success rates were highly correlated. Teenagers and parents met step and active-minute goals at least a third of the time, with parents meeting them more often. The step-count and active-minute success rates correlated significantly.
Rose and Soundy (2020) [98]	The integrative review and systematic search included studies on underprivileged children, with moderate to vigorous physical activity (MVPA)	N = 5886 Participants 9–16 years	The interventions included multi-sports, yoga, combined martial arts/group sports, dance, aerobic exercise, low-risk boxing, and muscle-building sports. The intensity was moderate to vigorous, the duration was 30 min to six hours, and the frequency was daily to weekly. Mainly noncompetitive group physical activity was included.	MVPA affects the mental health of disadvantaged children and adolescents, autonomy support promotes positive change, and age and gender affect the association. The review found that MVPA improves mental health and well-being in disadvantaged children and adolescents, but longitudinal studies and studies on specific types of physical activity are needed.
Bowling et al. (2021) [99]	The study design was a randomized controlled trial (RCT) with a pilot feasibility and acceptability trial design	N = 23 Participants 12–17 years	An Xbox One with Kinect motion sensor, a 12-week Xbox Live subscription, and three exergames (Just Dance 3, Shape Up 3, and Kinect Sports Season 2) were provided. Participants had 3 weekly exergaming sessions and 6 real-time telehealth coaching sessions.	This pilot study adapted and expanded an evidence-based exergaming and telehealth coaching intervention to improve PA, diet, video game play time, and sleep in youths with mental health and neurodevelopmental disorders and evaluated its feasibility and acceptability, including PA engagement. - PA improved and MVPA declined less after the intervention compared to controls. - The COVID-19 pandemic disrupted in-person learning and clinical care, making remote health behavior interventions for youth with NPDs a need. AGS may be a promising solution.
Van Woudenberg et al. (2020) [100]	Randomized controlled trial, clustered design, three groups, registered a priori in the Dutch Trial Registry, DANS archive	N = 446 Participants 9–16 years	Interventions for study participants: 1. Social network intervention: 15% of participants were asked to create physical activity vlogs. 2. Mass media intervention: Participants watched vlogs of unfamiliar peer. 3. All participants received a research smartphone and wrist accelerometer.	The social network intervention did not prove to be effective in increasing physical activity in adolescents, and no differences were observed between the social network intervention and mass media intervention. The study did not provide evidence that the social network intervention was more effective in increasing physical activity in adolescents compared to the mass media intervention or no intervention.
Zhou et al. (2018) [101]	The study design was a cluster randomized controlled trial (RCT) with a 2 × 2 factorial design	Participants 12–14 years	The study participants received the following interventions: Arm 1: School physical education intervention (SPE) with increased MVPA and VPA during school hours and provided nutrition education. - Arm 2: Afterschool program intervention (ASP) to increase MVPA and VPA after school and engage parents in supporting healthy habits at home via social media. Arm 3: SPE/ASP combined.	The study found that varying amounts of MVPA and VPA had an incremental effect on study outcomes using a 2 × 2 factorial design. It also addressed questions and debates surrounding school-based PA intervention research and incorporated current evidence and best practices to examine the effects of increasing PA doses in middle school students.
Kolb et al. (2021) [102]	Scoping review targeting systematic reviews and meta-analyses	Participants 6–17 years	-	The study identified physical health indicators for children and adolescents based on physical activity and sedentary behavior. Key study indicators included body composition, cardiometabolic biomarkers, fitness, harm and injury, and bone health. The study noted that sedentary behavior and health indicators are less consistent than physical activity.
Cadenas-Sanchez et al. (2021) [103]	Cross-sectional, longitudinal, and intervention designs were used in this systematic review and meta-analysis	Participants 2–18 years	-	Physical fitness was linked to youth mental health to a small to medium degree. Additional longitudinal and intervention studies are needed to prove causation. Cross-sectional studies predominated, indicating the need for more robust causation studies.
Schleider et al. (2019) [104]	Randomized controlled trial with three interventions, mixed-effects linear models, and using ANOVAs	N = 159 Participants 12–16 years	-	-
Agam-Bitton et al. (2018) [105]	The study design was a cluster randomized controlled trial, with a multi-site component	N = 259 Participants 13.3–14.3 years	Nine weekly 90 min sessions were led by female education or nutrition undergraduate or graduate students. The intervention encouraged adolescent self-esteem, positive body images, and media literacy. Overall, the mixed-gender group had higher body esteem, lower perceived current body image, and a smaller gap between current and ideal body images than the other two groups.	-
Jamnik and DiLalla (2019) [106]	Longitudinal design, multi-informant measures, prospective cohort study	N = 326 Participants 12–20 years	-	Preschool-aged internalizing problems are linked to adolescent health outcomes. More internalizing problems were linked to worse health outcomes regardless of age. Adolescent overeating and health issues were significantly predicted by parent-reported internalizing at age 5.
Gordon et al. (2020) [107]	Cluster RCT, intervention/control groups, block randomization, school-based social media literacy program evaluation	Participants 12–14 years	Australian secondary school grades 7–8 received a four-lesson social media literacy program. Participants were assessed at baseline, five weeks, six months, and 12 months later.	The study highlighted the negative impacts of social media on youths and outlines a novel approach using a school-based social media literacy program to mitigate these impacts.
Olive et al. (2019) [108]	A longitudinal cluster-randomized controlled trial was used	N = 821 Participants 7–12 years	The PE intervention was 4 years of specialized instruction. As a physical education program, the intervention’s frequency and dose were not specified.	The specialist-taught PE intervention initially decreased body dissatisfaction and depressive symptoms in children, but these effects were not sustained over the 4-year study period. Additionally, there was evidence of an overall increase in depressive symptoms over the 4 years for girls only.
Barkin et al. (2018) [109]	A randomized clinical trial with a parallel design and a controlled intervention and control group was conducted	N = 610 Participants 3–5 years	The intervention included a 36-month, family-based, community-centered program, with 12 weekly skills-building sessions, monthly coaching telephone calls for 9 months, and a 24-month sustainability phase providing cues to action.	The 36-month multicomponent behavioral intervention did not change BMI trajectory in underserved preschoolers in Nashville, Tennessee, compared to a control program. Intervention group children consumed fewer calories than control group children. Intervention parents took their kids to more community centers than control parents.
Bansal et al. (2021) [110]	Randomized, controlled, prospective, non-blinded, single-site, observational	N = 537 Participants 10–12 years	Each intervention group child received four monthly online one-on-one sessions from five standardized instructors for four months. To reduce screen time and promote self-care, the intervention targeted overweight and obese 10–12-year-old Indian pre-adolescents.	The pre-adolescents in the intervention group experienced weight loss, had reduced screen time and significant weight and obesity reductions in, and improved diet control.
Viggiano et al. (2018) [111]	Randomized controlled trial, pilot study	N = 1313 Participants 7–14 years	The intervention was playing Kaledo over 20 sessions.	Kaledo helped primary school students (7–11 years old) learn about nutrition and change their diets. At 8 months, the treated group had a lower BMI z-score, were participating in more exercise, and a better diet.

## 5. Discussion

One study’s main findings [1] suggest that the gamified intervention had positive effects and may play an essential role in encouraging adolescents to participate in physical activity. This study also highlighted the alignment of gamified Behavior Change Techniques with core concepts of Self-Determination Theory and the need to tailor game elements for specific populations. Another gamification application implemented in another study [80], HelperFriend, was feasible and acceptable for young children and has shown potential to improve knowledge, healthy behavior intention, and dietary intake. However, further well-powered randomized controlled trials are needed to establish the effectiveness of these applications. In another study, a mobile health game application [81] was positively perceived by participants, with high usability and acceptability ratings. In addition, there was a statistically significant reduction in general health problems and a trend toward a reduction in depression and anxiety symptoms during the intervention period.

Another study [88], aiming to evaluate the effectiveness of a smartphone application with a gamification approach to promote healthy lifestyles and weight reduction in overweight and obese adolescents, highlighted the medium-term effects of gamification-based interventions for controlling weight in adolescents, providing insights for future research. The results of the proposed trial can guide clinical practice. They can be used by other groups aiming to evaluate smartphone applications for managing obesity and other chronic metabolic diseases in adolescents.

The gamification strategy appears to prevent childhood obesity and reduce systolic blood pressure in school children in Santiago, as pointed out in another study [89]. Active video games, studied in a related research protocol involving children [95], led to favorable changes in weight, body mass index, reaction times, and self-concept in inactive and technologically engaged children. A qualitative analysis showed high levels of enjoyment in the active video game group, suggesting a motivation to continue playing. The study revealed diverse contributions of active video games to children’s physical, social, intellectual, and personal development.

The primary study findings regarding physical activity and eating habits [110] include positive effects in the intervention group, such as weight loss and reduced screen time, significant reductions in weight and obesity among preadolescents, and increased dietary control in the intervention group. This study highlights the importance of understanding adolescents’ dietary intake and nutritional status because of the critical physical and psychological changes during this period and the potential long-term effects of health-related behaviors and conditions during adolescence. The findings highlight the importance of addressing adolescents’ dietary and nutritional needs to promote their future health. Another gamification intervention, reported in [91], led to significant increases in physical activity, mainly when participants had selected immediate goals. No other gamification arm had consistent increases in physical activity compared to the control arm. According to the research results, motor skill intervention can enhance the development of all children—students who completed more quizzes performed better on subsequent tests, supporting these results.

Gamified quizzes led to significantly better scores on the first test, but this effect was not due to more quizzes being completed in the gaming group. The beneficial effect of gamification did not persist for subsequent tests, and higher-performing students benefited more from gamification than lower-performing students. Another study demonstrated [81] the potential of a gaming environment to provide comprehensive supportive care to pediatric patients with cancer and other chronic conditions. In this study, the utility of gamification methods was tested in both typically developing children and adolescents and in clinical populations. A corresponding study [86] found positive results with improved health habits, high motivation from conducting sessions (face-to-face) using gamification strategies, and enhanced patient perception of physical activity.

Gamification interventions to enhance physical activity can improve adolescent mental health, as reported in [83], but more research is needed to confirm the effects on children’s mental health. Promoting physical activity and reducing sedentary behavior can protect the mental health of children and adolescents. Longitudinal and cross-sectional studies have shown significant associations between physical activity and lower levels of psychological illness, as well as greater psychological well-being, as well as between sedentary behavior, increased psychological illness, and lower psychological well-being. Another study that examined groups of children (experimental group and control group) through gamification methods regarding their knowledge about healthy eating and adherence to the Mediterranean diet demonstrated better compliance of the experimental group than the control group. Nevertheless, after the experimental design, a significant improvement was observed in the knowledge of the two groups about healthy eating after the end of the research program, highlighting the need for continuous education of children and adolescents in matters of physical health promotion.

Another study’s main findings [86] included significant improvements in children’s knowledge of healthy eating and adherence to the Mediterranean diet, with the experimental group showing better adherence than the control group. The study also observed a significant improvement in both groups’ knowledge of healthy eating, reaching higher levels in the long term than in the two previous assessments.

According to [90], gamification can improve learning outcomes in health professions’ education, especially when using game features that improve learning behaviors and attitudes toward learning. Most included studies conducted in the US and Canada used digital technologies to develop and implement gamified teaching and learning. No adverse effects were observed from the use of gamification. Almost all interventions included evaluative play features, mostly in combination with conflict/challenge features. This combination of features, in particular, has been found to increase the use of learning materials, sometimes leading to improved learning outcomes.

Kaledo, an electronic health game, was shown to be effective in improving nutritional knowledge and modifying dietary behavior in younger students (7–11 years) attending primary school [111]. The treated group showed a significantly lower body mass index z-score at eight months, increased physical activity, and improved eating habits. The mobile health game was positively perceived by patients, leading to continued engagement, and the intervention showed potential effectiveness in improving health outcomes for pediatric patients with chronic diseases. The study demonstrated the potential of a gaming environment to deliver comprehensive supportive care, indicating a broader applicability to chronic conditions.

Following an analysis of the publications in the results section, we will now discuss the extracted results and their conclusions. Most analyzed studies concluded that innovative health promotion initiatives are in demand as children and adolescents experience more physical and mental health difficulties [112]. Gamified situations could make health practices more exciting and engaging [113]. Gamified scenarios can promote many healthy behaviors. Exergames and fitness apps make exercise fun, encouraging regular exercise. Nutrition-focused games and gamified challenges or tracking systems can teach kids and teens about healthy eating [114].

Furthermore, gamified education can enhance sleep, dental, and safety practices. They make these activities more desirable and lucrative, encouraging regular repetition. Furthermore, gamification can also promote mental health. Coping, emotional regulation, social skills, and mindfulness games can help kids and teens maintain mental health. Growth mindset games can boost resilience and motivation, improving mental health [115].

Additionally, gamified interventions can teach children and adolescents about mental health issues and encourage empathy and understanding, promoting mental health literacy and reducing stigma. Also, gamified situations are promising but pose hazards and obstacles. Screen time and games encouraging competition over cooperation may harm children’s and teenagers’ health.

Another essential element for promoting mental health is game design, which can affect the results. For instance, uninteresting or unchallenging games may not encourage wellness. To sum up, gamified health promotion for children and adolescents is promising and merits additional study. To ensure efficacy, safety, and benefit, multidisciplinary teams of health professionals, educators, game designers, and psychologists should develop these interventions and design policies [116,117].

The discussed articles highlighted several research findings regarding the advantages of gamification techniques in enhancing children’s and adolescents’ physical and mental well-being. The systematic review found that most studies employing gamification primarily concentrated on encouraging physical activity and healthy eating while paying less attention to promoting mental health. Specifically, there is limited focus on addressing interpersonal relationships, sex education, gender differences, internet addiction, stereotypes, community stigma, safety, and accident prevention, as outlined in this systematic review. Moreover, certain studies exhibit contradictory findings regarding gamification techniques, which merit careful consideration. Implementing research protocols that utilize gamification techniques to address physical activity and nutrition and raise awareness among children and adolescents would be highly advantageous in promoting the mental well-being of this vulnerable and expanding demographic.

### Future Implications

According to the review of the literature on promoting the physical and mental health of children and adolescents through gamification techniques and game scenarios, the primary focus of all studies has been on enhancing eating and physical activity habits. Future research could benefit from a protocol that combines and includes other parameters mentioned in this review, such as accident prevention, internet addiction, sex education, stigma and stereotypes, interpersonal relationships, gender differences, and the parallel use of game scenarios for the optimal awareness and familiarization of children and teachers in these matters.

It is important to note that one of the primary policies in all European Union (EU) countries is the advocacy for health education among children. Health education enhances students’ understanding, proficiency, and favorable dispositions toward health. Health education provides instruction on various aspects of well-being, including physical, mental, emotional, and social health. It serves as a source of motivation for students to enhance and sustain their physical well-being, prevent illnesses, and mitigate hazardous behaviors. To address this, primary school teachers must employ inventive approaches to facilitate the comprehension of health-related matters, self-defense, and public health among children. The COVID-19 pandemic exposed the insufficient health education provided to children in multiple countries and the inadequacy of digital tools for remote learning. Digital Game-Based Learning (DGBL) is a cutting-edge approach that aids children in learning in traditional classroom settings and through remote means. Digital Game-Based Learning (DGBL) is an effective method for educating primary school students, as it fosters a fresh approach to the learning environment that aligns with the students’ interests. Digital Educational Games (DEGs) are a novel advancement in primary education that can augment children’s learning and skill acquisition. Incorporating Digital Education Games (DEGs) into the primary education setting can significantly enhance the educational system’s reform efforts. Primary education teachers can play a vital role in supporting children’s Digital Game-Based Learning (DGBL). Teachers can offer high-quality and inclusive education through online platforms by implementing digital tools and methods. This enables them to provide comprehensive training to children, including important topics such as health issue prevention. The authors of this article are participating in the execution of a European project focused on creating visual educational games with a specific objective. These games aim to assist primary school teachers in delivering health education to children aged 6–12. Teachers will receive training in theoretical concepts and the practical implementation of the DGBL method. This will involve utilizing the Digital Educational Games created by the project partners. The primary goals of the project are to:Facilitate the teachers (and children) in cultivating a scientific perspective on health, encompassing both traditional and modern health concepts;Empower teachers to deliver top-notch health education and employ curricula that embody the attributes of effective health instruction;Advocate for integrating Digital Game-Based Learning (DGBL) in primary schools.

The outcomes of the project above will be disclosed after its conclusion.

Future studies should investigate the long-term effectiveness and durability of gamification interventions, emphasizing incorporating diverse and inclusive design principles to address the needs of underserved populations. It is crucial to address the limitations identified in the study, including potential overstimulation, addiction to gamified activities, equity in digital access, and privacy concerns, to make progress in this field. Moreover, long-term studies and robust randomized controlled trials must be conducted to analyze the cause-and-effect relationship between gamification and health outcomes. Researchers could further investigate the effects of gamified interventions in different demographic groups and in various contexts. This analysis would examine how these interventions can effectively address cultural sensitivities and literacy levels while avoiding the potential negative consequences of reinforcing stereotypes or stigmatizing individuals. Collaboration among diverse teams of healthcare professionals, educators, psychologists, and game designers can create innovative, ethically designed, gamified solutions that are easily accessible, user-friendly, and effectively integrated into daily health routines. Future research should address the methodological gaps, including establishing standardized evaluation frameworks, comprehensive risk–bias assessments, consideration of publication bias, and the systematic organization of data across various studies. These efforts will strengthen the evidence supporting the use of gamification to promote physical and mental health in children and adolescents.

## 6. Conclusions

To summarize, the current article has highlighted the potential of gamification and its complex mechanisms in fostering the well-being of children and adolescents, both physically and mentally. Our investigation highlighted the significant capacity of interactive and captivating game mechanics to promote substantial changes in behavior, bolster emotional resilience, and ultimately improve health outcomes for young individuals. Therefore, gamification is a cutting-edge and persuasive health education and promotion method in an ever-growing digital era.

Nevertheless, gamification presents specific difficulties and restrictions, encompassing hazards such as excessive stimulation, reduced inherent drive, and apprehensions regarding unequal participation among users. These potential drawbacks highlight the significance of meticulous, age-appropriate, and comprehensive designs for gamified interventions. Furthermore, our analysis has emphasized the necessity of further research to thoroughly evaluate gamified methods’ long-term efficacy and durability, specifically in digital literacy, internet safety, and sexual education, where the medium holds considerable influence and accountability.

Based on numerous empirical studies, we have observed favorable effects in various health-related contexts, including enhanced dietary habits and physical fitness routines, improved mental health, and training in social skills. The empirical evidence also indicates that although gamification is effective in short-term interventions, conducting more rigorous, well-planned, randomized controlled trials to confirm these findings over longer durations and in various environments is crucial.

For these interventions to be successful, it is crucial to develop them in a way that incorporates knowledge from various fields such as psychology, education, technology, and healthcare. This will result in gamified experiences that are both engaging and effective while also ensuring safety. The studies discussed in this paper support the adoption of gamification in a future-oriented manner while also emphasizing the need for a combination of traditional approaches, an understanding of the cultural background of the target demographic, and a solid dedication to ethical considerations in the design and implementation process.

Based on the extensive evidence and discussions presented, this paper acknowledges the significant advantages of gamification in promoting physical and mental well-being. The widespread presence of technology and its ever-changing role in our everyday lives implies that gamification will likely have a growing impact on health-related education and behavior modification, particularly among younger demographics. However, as we explore this hopeful new area, it is essential that we continuously evaluate and improve these methods. They should be based on solid evidence, customized to meet the various requirements of young users, and seamlessly incorporated into comprehensive health promotion strategies.

To progress, we need to carefully explore the complex field of gamification, promoting its advancement through interdisciplinary cooperation, continuous research, and unwavering commitment to young people’s physical and mental welfare. By making these coordinated endeavors, we can fully exploit the potential of gamification and guarantee that future digital environments can promote young individuals’ well-being, involvement, and adaptability.

## Figures and Tables

**Figure 1 behavsci-14-00102-f001:**
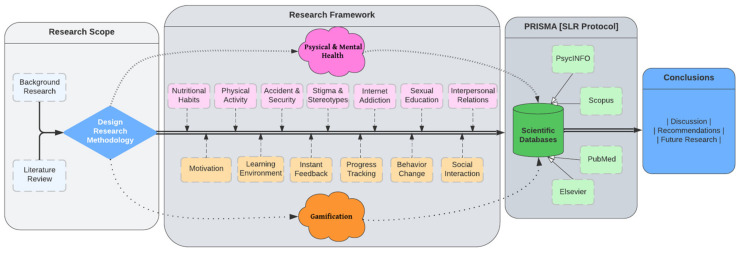
Flowchart of research framework.

**Figure 2 behavsci-14-00102-f002:**
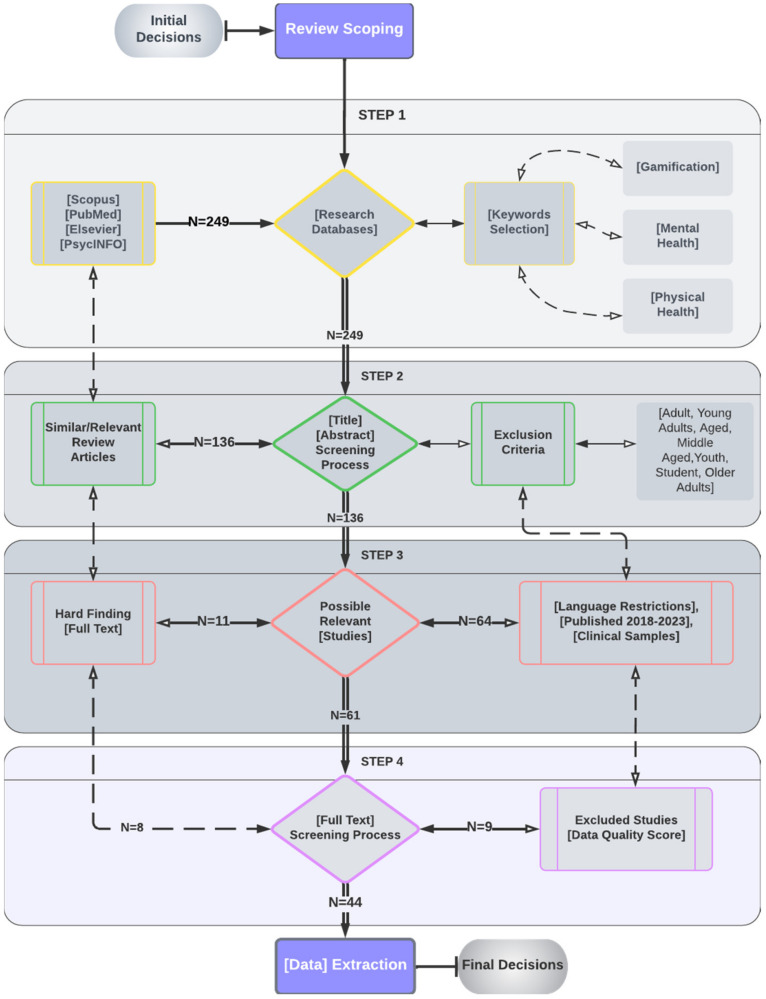
PRISMA methodology flowchart.

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
