# Peer review of "Promoting Physical and Mental Health among Children and Adolescents via Gamification—A Conceptual Systematic Review"

_behavsci, 2024, doi:10.3390/bs14020102_

Round 1

Reviewer 1 Report

Comments and Suggestions for Authors

A Conceptual Systematic Review embeds a hybrid approach to content analysis in the larger framework of a systematic review (Schreiber and Cramer, 2022). When striving for untangled and clearly defined terms in educational research, the combination of these two methods utilizes their respective advantages and allows their shortcomings to be compensated by the other method (Khirfan et al., 2020). The review protocol used to conduct the systematic review we are evaluating follows the PRISMA method which is a success and attests to the good work done by the authors. 

Author Response

Dear Reviewer,

We appreciate your insightful feedback. We have thoroughly revised our article according to the suggestions you provided, and all of the modifications have been highlighted in grey.

Reviewer 2 Report

Comments and Suggestions for Authors

Unfortunately, this article does not bring anything out of the ordinary in terms of its originality. The work does not present adequate new and significant information to justify publication, because although the Introduction section offers a preliminary understanding of the topic and conveys some insights into the current state of knowledge in the field of study, the authors need to thoroughly outline a gap in existing research. 

The introduction would benefit from presenting a structure of the manuscript.

The literature review lacks a clear direction, as the paragraphs are poorly connected from a logical perspective. It is recommended to use logical bridges or sentences that link these parts together to improve the cohesion of the literature review paragraphs.

In a chapter reviewing the state of the art, authors must interconnect facts found by studies. This does not happen in this literature review. The authors make scientific statements and assume assumptions without pointing to evidence or studies for this. This happens, for example, between line 98 and line 110, or between lines 149 and 163, or 172 and 177, or 333 and 352. This is not at all recommended in the literary review chapter.

Figure 1 and Figure 2 are out of place.

Chapter 3 is incomplete. It does not display all the data from the query performed. The process that appears in figure 2 is not described and table 1 is not even referenced in the text.

Some of the conclusions that appear in section 5 had already been identified in section 2 of the literary review.

This article presents several problems, some of them too serious to occur in an indexed journal. Furthermore, the article also demonstrates a lack of care and panache, such as out-of-place figures and the table without being referenced. The conclusions are poor and there is no more in-depth comprehensive analysis.

Author Response

(The authors gave the same response as above.)

Reviewer 3 Report

Comments and Suggestions for Authors

Thank you for submitting your article to the Behavioral Sciences journal. The article presents several commendable attributes. It addresses a highly relevant and timely topic, intersecting digital technology with health promotion in youth, which I truly believe is crucial in today's evolving technological landscape. I also like the thoroughness of the literature review as it encompasses a wide range of studies and sources and adds a robust depth to the research. However, as I will point out later, the discussion is not as thorough as this section. Here are some key points that may improve the article:

INTRODUCTION

1. The introduction does not clearly identify the existing gaps in the literature that this systematic review aims to address. Highlighting these gaps would underscore the significance and need for this study. For instance, what is the difference of this study compared to existing ones (e.g., DOI: 10.1007/s11920-023-01453-5)?

2. There's no mention of the technological prerequisites or the potential disparities in access to digital devices among children, which can influence the effectiveness and reach of gamification.

3. It would be helpful to offer clear definitions for essential terms early in the paper. Specifically, clarifying what "Children and Adolescents" refers to in the context of your study is important. Does this term encompass a specific age range? Such definitions would enhance the clarity and precision of your research, aiding readers in better understanding the scope and applicability of your findings.

LITERATURE REVIEW

4. While the benefits of gamification are extensively discussed, there's a noticeable absence of potential drawbacks or challenges associated with this approach, which would offer a more balanced view.

5. While this section provides many benefits of gamification, providing a brief history or evolution of gamification among certain age ranges might be helpful. This can offer readers a perspective on how the field has evolved.

MATERIALS AND METHODS

6. The terms "gamification", "game-based learning", and "video games" are often used interchangeably in the literature. The author must clarify how they distinguished between these terms in their study. Did they consider them synonymous, or did they differentiate between them based on certain criteria? For instance, Table 1 Row 2 is about an intervention program through a video game

Providing a clear definition or operationalization of these terms would ensure consistency in study inclusion and offer readers a clearer understanding of the scope of the review.

7. There's no mention of how the quality or risk of bias of the included studies was assessed. This is crucial in systematic reviews to ensure the reliability and validity of the findings. Include a risk of bias assessment for the included studies. 

8. While the data extraction process is mentioned, there's no detail on how the data from different studies were synthesized, especially if there were variations in the measures or outcomes reported in the original studies. Provide more information on the data synthesis process, especially if there were variations in the measures or outcomes among the included studies.

9. There's no mention of whether publication bias was assessed. Techniques like funnel plots or Egger's regression test can help identify potential biases in the included studies.

DISCUSSION AND CONCLUSIONS

10. The subheading includes the term "Conclusions" yet there is another section for "Conclusions". Did you mean Discussion and Future Implications?

11. Given the nature of a systematic review, the discussion is relatively brief. Expanding this section to provide a more in-depth analysis of the findings would be beneficial. This could include a deeper exploration of how the results align with or differ from existing literature, potential reasons for these findings, and implications for future research and practice.

12. While the discussion highlights the potential benefits of gamification in health promotion, it could benefit from a more critical analysis. This might include exploring the limitations and challenges of the studies reviewed, such as methodological weaknesses or inconsistencies in findings.

13. The discussion would be enhanced by placing the findings within a broader context. This could involve discussing how these findings relate to broader trends in health promotion, technology use among children and adolescents, or societal changes.

14. The discussion briefly mentions hazards and obstacles related to gamification, like increased screen time and competitive elements. Expanding on these points would provide a more balanced view. Discussing how these challenges might be mitigated or what future research is needed to address them would be valuable.

15. While the discussion suggests that multidisciplinary teams should develop interventions, elaborating on specific policy implications or practical recommendations for different stakeholders (educators, health professionals, game designers) would make the discussion more actionable.

16. The discussion could benefit from comparing gamified interventions with other types of health promotion strategies. This would provide a more comprehensive understanding of where gamification stands in the broader landscape of health interventions for children and adolescents.

17. The discussion could also touch upon how these gamified health interventions can be made accessible and relevant to diverse populations, considering cultural, socio-economic, and individual differences among children and adolescents.

CONCLUSIONS

18. While the conclusion effectively summarizes the potential of health promotion gamification, it could benefit from more depth and detail. Elaborating on the specific ways in which gamification can improve health, or citing key findings from the review, would add more substance to the conclusion.

19. The mention of drawbacks is important, but these could be more explicitly detailed. Discussing specific limitations and challenges of gamification in health promotion, and suggesting ways to address them, would provide a more balanced and realistic perspective.

20. The call for studying long-term effects and health benefits is valuable, but the conclusion could further suggest specific areas or questions for future research, based on the findings of the review.

21. Expanding on the implications for policy makers, health practitioners, and game developers would make the conclusion more actionable. Specific recommendations or guidelines for these stakeholders based on your findings could be highly beneficial.

22. The statement that gamification should be part of broader health promotion programs is key. Further elaboration on how gamification can be integrated with other health strategies and practices would provide a clearer roadmap for implementation.

23. Including a note on the importance of tailoring gamified interventions to diverse populations, considering cultural, socio-economic, and individual differences, would add an important dimension to the conclusion.

24. A brief reflection on the current state of research in this field, based on your findings, would provide context and highlight the contribution of your review to the existing literature.

Comments on the Quality of English Language

Moderate editing of English language required

Author Response

(The authors gave the same response as above.)

Reviewer 4 Report

Comments and Suggestions for Authors

First of all, I would suggest changing the title: Promoting Physical and Mental Health Among Children and Adolescents via Gamification. A Conceptual Systematic Review // or Promoting children and adolescents' physical and mental health through gamification. A Conceptual Systematic Review

Then I have detected in the Literature review a lot of statements you make which should be referenced. Please see below just a few examples:

Lines 68-69: Reference needed here and in the next paragraph as well: "Gamification, which integrates game mechanics and design into non-game environments to increase engagement, motivation, and overall enjoyment, is one such strategy garnering significant traction."

Lines 101-106: Reference needed (how do you know all this information?): "Gamification can make health-related behaviors, such as physical activity, healthy eating, and maintaining excellent hygiene, more appealing in health promotion. Children and adolescents can be motivated to adopt healthier lifestyles by using mobile applications that transform physical activity into a game or award points for healthy meals, for instance. Additionally, gamification can be utilized to improve mental health. Games that teach coping skills, emotion regulation, mindfulness, and social skills can equip children and adolescents with the tools necessary to maintain mental health."

Line 197: (Lieberman, 2001) - turn this into the appropriate style of referencing

Lines 276-278: Are you sure [33] is the correct reference here? "It is possible for gamified interventions to be effective in combating stereotypes. In a  study conducted [33], researchers utilized an avatar-based game to challenge adolescents' gender stereotypes. The results indicated a significant decline in gender stereotyping and an increase in gender bias awareness."

Line 603: According to a review... (your review?)

After you check and adapt the referencing, please note that the Literature review is more than half the paper. I think you should restructure it in such a way that it is more compact. Maybe also number the subsections (2.1, 2.2 etc).

About the Methodology: it is very briefly described and therefore I find it incomplete. Very importantly: how many articles did you analyze? How many of these (total number of articles) were about each category (i.e. nutrition, physical activity, and so on)? How many of the articles dealt with 2 (or more) of the categories? Perhaps you can add a column to Table 1 in which you mention the topic/category. 

You touch upon certain Limitations of using gamification (lines 564-566 and 596-598) but I think you could create a separate paragraph/fragment clearly stating that these are limits and they need to be carefully considered when implementing this method. 

Overall I believe your paper has some strengths (interesting and engaging topic, Table 1), but there are parts which can be improved (see above). 

Author Response

(The authors gave the same response as above.)

Round 2

Reviewer 2 Report

Comments and Suggestions for Authors

Despite the authors saying that they revised the article according to my instructions. They do not convincingly indicate how they did this. In fact, they responded in the same way to all reviewers. Just as you expect insightful feedback from reviewers, the opposite should also happen. Authors must indicate how they responded to reviewers' concerns.

Furthermore, some of the problems highlighted in the first review were not corrected. For example: Figure 1 and Figure 2 are still out of place (Figure 1 is referenced on page 3 and appears on page 11 and figure 2, which describes a method that belongs to section 3, continues to appear in section 4.

The literature review still lacks a clear direction, as the paragraphs are poorly connected from a logical perspective. 

Author Response

Dear Reviewer,

Thank you for your comments. The manuscript has been revised to address the changes as indicated. Changes in the literature review are indicated in yellow. Paragraphs are connected and clarified using logical bridges and sentences to improve their cohesion. Figures 1 & 2 are placed on the correct page/section.

Thank you for your valuable comments and constructive collaboration.

Evgenia Gkintoni 

Reviewer 3 Report

Comments and Suggestions for Authors

Congratulations!

Comments on the Quality of English Language

Proofread to ensure that there's no mistake prior to publication.

Author Response

Thank you for your valuable comments on the improvement of our manuscript and the constructive collaboration.

Reviewer 4 Report

Comments and Suggestions for Authors

Dear Authors,

Thanks for reworking your paper and incorporating my suggestions.

Author Response

Thank you for your valuable comments on the improvement of our manuscript and your constructive collaboration.